# Robust Spike-based Decoupled Federated Information Bottleneck Learning with Spiking Neural Network under System Heterogeneity

## Abstract

As embedded devices become increasingly prevalent in intelligent systems, low-power system in resource-constrained environments has emerged as a key challenge. Spiking neural networks (SNNs), with their sparse and event-driven computation, have shown great potential as a low-power candidate for embedded devices. In federated learning scenarios, where multiple energy-constrained devices collaborate, adopting efficient SNN models with effective training methods is critical. However, research on training SNNs within federated learning systems is still very limited, particularly in terms of how to achieve both energy efficiency and robustness under system heterogeneity. This gap presents a significant opportunity for further exploration of SNNs in distributed learning settings. In this paper, we investigate a significant and innovative problem in robust spike-based federated learning, particularly in the presence of noise, and system heterogeneity. We majorly consider two types of system heterogeneity in this study, including data and client participation heterogeneity. To address this, we propose a novel federated learning framework, spike-based decoupled federated information-bottleneck learning (SDFIL), to enable robust, low-power federated learning through SNNs under system heterogeneity. Specifically, we design a decoupled information bottleneck principle tailored for local SNN training to maximize the mutual information between ground truth and model predictions while minimizing mutual information between intermediate representations. This method effectively minimizes the impact of outliers in non-independent and identically distributed (non-IID) data on model updates, thereby enhancing the performance of federated SNNs, resulting in enhanced robustness and reduced sensitivity to outliers. We evaluate the proposed SDFIL algorithm across a variety of settings, including different noise levels and varying degrees of system heterogeneity. The experimental results indicate that SDFIL demonstrates superior robustness compared to competing methods and generally achieves an improvement in overall accuracy of 5% to 10%. Additionally, it can achieve up to 7.7× higher energy efficiency compared to traditional artificial neural networks (ANNs).

## 1 Introduction

Neuromorphic computing represents a groundbreaking approach to achieving artificial general intelligence, modeled after the brain's processing mechanisms (Renner et al., 2024). Its event-based processing provides several advantages, including low power consumption, low latency, and high biological plausibility (Frenkel et al., 2023). With advancements in neuromorphic chips, such as TrueNorth, Loihi, and Tianjic, neuromorphic computing has found applications across various fields (Akopyan et al., 2015; Davies et al., 2018; Pei et al., 2019). A prominent model for neuromorphic computing is the spiking neural network (SNN), which employs spiking neurons as its fundamental unit (Meng et al., 2023; Fang et al., 2022; Yao et al., 2023). SNNs not only exhibit low-power efficiency but also achieve performance on par with deep neural networks (Qiu et al., 2023; Kim et al., 2019; She et al., 2022). Due to this power efficiency, SNNs show great potential for deployment in embedded devices (Kucik & Meoni, 2021; Ottati et al., 2023). However, embedded devices often require continuous data collection for training and updates.

Local clients typically have limited private data and lack generalization ability (Mendieta et al., 2022; Nguyen et al., 2022). However, due to data silos and privacy concerns, centralized learning methods are impractical in real-world applications (Yin et al., 2021). To address this, federated learning has been introduced as a distributed machine learning framework (Zhang et al., 2021). It enables numerous clients to collaboratively train a model with decentralized data, ensuring that private data is not shared with a central server, thus safeguarding data privacy (Zhuang et al., 2021). Although federated learning plays a crucial role in large-scale, distributed edge learning systems that prioritize privacy, current research on SNNs has yet to develop extensive methods for federated learning within a distributed training framework.

When applying federated learning to SNNs, three key challenges arise, as shown in Figure 1. First, data heterogeneity among clients is particularly problematic for SNNs. Clients typically generate non-independent and non-identically distributed (non-IID) data, which disrupts the common IID assumption (Besbes et al., 2024; Morafah et al., 2022). As the number of clients increases, managing the impact of this heterogeneity on SNN training becomes even more difficult, leading to a critical challenge for federated learning with SNNs. Second, to mitigate the risk of data leakage through model updates, some approaches introduce noise to obscure gradients (Li et al., 2022). Thus, federated learning must ensure protection against gradient noise during model updates. Third, due to variations in device performance and network connectivity, not all clients may participate in each training iteration, resulting in client participation heterogeneity (Cui et al., 2022). Given that SNNs are more computationally intensive due to event-driven processing and temporal characteristics, this heterogeneity in client participation exacerbates the challenge of maintaining model consistency and accuracy across a distributed system. In this study, data and client participation heterogeneity are collectively referred to as system heterogeneity. Addressing system heterogeneity is critical for ensuring the robustness and scalability of federated learning in SNN applications. Unlike previous

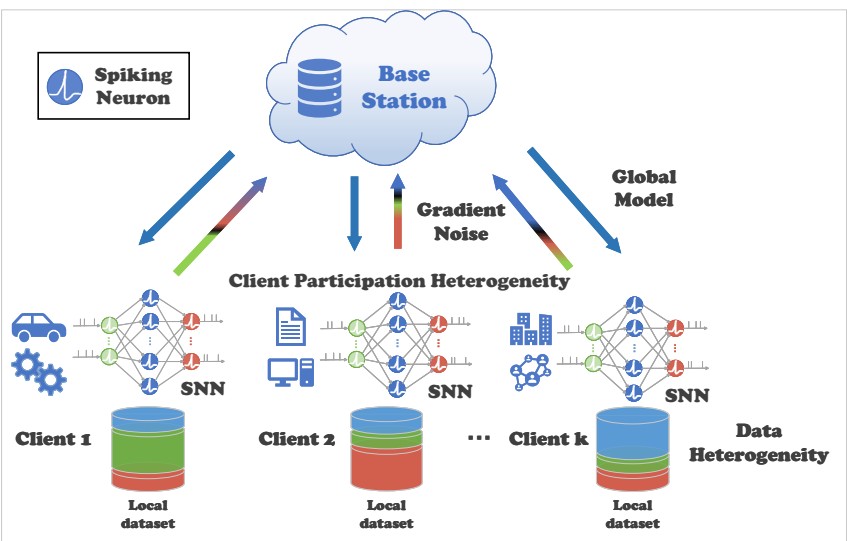

Figure 1: Illustration of spike-based federated learning based on SNNs with gradient noise, client participation heterogeneity, and data heterogeneity.

studies that focus solely on efficient SNN training, we approach SNN training from the perspective of federated learning. To address the aforementioned challenges in federated learning with SNNs, we propose a novel algorithm called spike-based decoupled federated information-bottleneck learning (SDFIL). Our approach is aiming to achieve robust, low-power federated learning in the presence of noise, and system heterogeneity. The contribution of our work includes three folds:

- We study a novel and important robust spike-based federated learning problem with gradient noise and system heterogeneity that includes data and client participation heterogeneity, and design a novel information bottleneck based optimization approach.

- We derive a novel decoupled information bottleneck algorithm for spike-based federated learning. It is useful to minimize the effects of outlier of non-IID data on model updating, which can improve the robustness and scalability of federated SNNs.

- We test the proposed SDFIL algorithm on various type of settings, including different noise strength and different levels of system heterogeneity. Experimental results demonstrate the better robustness of SDFIL than competing methods and generally achieves an improvement in overall accuracy of 5% to 10%.

## 2 BACKGROUND

### 2.1 FEDERATED LEARNING

Federated learning refers to a distributed training method that has significant data privacy advantages and lower communication cost compared to centralized training models (Bohte, 2011). This framework consists of a central server and $K$ edge clients capable of processing data independently. Each of the clients $k = 0, 1, 2, \ldots, K$, has its own local dataset $D_k = \{(x_{k_i}, y_{k_i})\}_{i=1}^{n_k}$, with $|D_k| = n_k$. The client split is denoted by $K/P$, where $K$ is the total number of clients and $P$ is the participating devices per round. At round $r$, each selected client that constitutes the set of training participants $C_r$, with $|C_r| = P$, uses their private local data to calculate the average gradient $\Delta W_k^r$ in the current local model, and sends it to the central server. The central server then aggregates these gradients and updates the global model:

$$W^{(r+1)} = W^r + \gamma \sum_{k \in C_r} \frac{n_k}{n} \Delta W_k^r, \tag{1}$$

where $n = \sum_{k \in C_r} n_k$ represents the total number of samples that participated in the training at round $r$. The learning rate $\gamma$ represents the magnitude of the weight change. federated learning constructs a realistic scenario, where the model can be trained on different types of edge devices that typically have lower power consumption and limited compute and memory resources. Therefore there has been a great deal of interest in energy-efficient federated learning based on SNN (Tumpa et al., 2023).

### 2.2 SPIKING NEURAL NETWORKS

SNNs are brain-inspired computational models that transmit information through discrete events known as spikes. In this context, we employ the widely recognized leaky integrate-and-fire (LIF) neuron model as the constituent unit of SNNs due to its simplicity and scalability.

During forward propagation, the LIF neuron utilizes the membrane potential as its internal state. The input spikes from presynaptic neurons are weighted and aggregated, contributing to the membrane potential. When this potential reaches a certain threshold, the neuron generates a spike as the output signal and reset its state to the resting potential. The discrete-time form of LIF neurons is as follows:

$$u_i^t = \lambda u_i^{t-1} + \sum_{j \in N} w_{ij} s_j^{t-1}, \tag{2}$$

$$s_j^{t-1} = \begin{cases} 1, & \text{if } u_j^{t-1} > \theta \\ 0, & \text{otherwise} \end{cases}, \tag{3}$$

where $u_i^t$ is the neuron membrane potential at time step $t$, and $\lambda$ is the membrane potential leak. $w_{ij}$ represents the synaptic weight between neuron $i$ and $j$, and $\theta$ is the threshold voltage value. $s_j^{t-1}$ denotes the spike output of the neuron $j$, which is a binary function. When $u_i^{t-1}$ reaches the threshold $\theta$ at the time step $t$, the neuron $j$ generates a spike and takes value 1 as output and 0 otherwise. In this study, we employ the most commonly used error-driven global training method, back propagation through time (BPTT) which is designed to propagate errors across multiple time steps. The gradients of global loss are computed by backpropagating the errors through the unrolled network, which can be calculated as accumulating the gradients at each time step:

$$w_{ij} = w_{ij} - \eta \Delta w_{ij}, \tag{4}$$

$$\Delta w_{ij} = \left( \frac{\partial L}{\partial w_{ij}} \right) = \sum_{t=1}^{T} \left( \frac{\partial L}{\partial s_i^t} \frac{\partial s_i^t}{\partial u_i^t} \frac{\partial u_i^t}{\partial w_{ij}} \right), \tag{5}$$

where $L$ is the target loss function. The activation function $s_j^{t-1}$ is a Heaviside function and cannot be differentiated at the threshold. To solve this challenge, Neftci et al. (2019) proposed a smooth surrogate function for computing the gradient of the activation function. This approach allows SNNs to be trained using conventional methods while still enabling the inference of spiking signals during forward propagation.

## 3 METHOD

### 3.1 PROBLEM DEFINITION

In our investigation of federated learning frameworks, we methodically address three predominant challenges: client participation heterogeneity, data heterogeneity, and gradient noise.

In federated learning, client participation heterogeneity poses a significant challenge to the scalability. This issue mainly arises when the amount of training data $N_{\text{train}}$ remains constant but the number of clients $K$ increases, which reduces the amount of private data $n_k$ on each client, adversely affecting the data diversity in each training round. Although the total number of clients represents one dimension of scalability, the proportion of participating clients in each round, $K/P$, is also crucial to performance, especially under non-IID datasets. Furthermore, data heterogeneity and privacy concerns complicate model training and scalability. Individual clients, varying in geographic, demographic, or behavioral factors, may have unique data distributions $P_k$, often with skewed category $\Sigma$ representation $D_k \sim P_k(\Sigma)$, which challenges the global model's ability to learn uniformly from all clients. Here, $\alpha$ serves as a parameter that dictates how unevenly samples are distributed among users. A smaller $\alpha$ value leads to more uneven (non-IID) distributions. Moreover, while the base station $B$ updates the global model using gradients from clients, it cannot access local data directly. This arrangement, although protective, doesn't fully prevent potential data inference from exchanged gradients. To enhance data privacy, gradients sent to the base station are typically obfuscated with noise, addressing the risk of data reconstruction. Clearly, our aims are the following: by minimizing the loss on local datasets, the resulting gradients should exhibit high sample efficiency and maintain good generalization capabilities, while also retaining information even after gradient noise is introduced.

### 3.2 PROPOSED SDFIL

Although previous work has presented the concept of information bottleneck for SNNs, they have not considered decoupling feature to solve overfitting and improve generalization ability. Thus, we propose to build a loss function that comprise the sufficiency of information bottleneck, compressed statistical amount and decoupling feature in federated learning of local model. Since it is challenging to design this kind of loss function, we rebuild the variational lower bound of information bottleneck and apply it in the spiking federated learning framework (Venkatesha et al., 2021). The loss function $L_j(w)$ of the $j$th client is defined as

$$L_j(w) = \frac{1}{N_j} \sum_{i \in M_j} l(w, i), \tag{6}$$

where $N_j$ represents the data sample number of the $j$th client, and $l(w, i)$ represents the loss function of the $i$th sample. $F_j$ represents the collection of the data index, with the length $N_j$. In the conventional federate learning algorithm, the cross-entropy loss function is used for training for providing the sufficient representation of input, which is formulated as

$$l(w, (X, Y)) = -\sum_{u=1}^{r} Y_u \log(Y^u), \tag{7}$$

where $Y$ represents the one-hot encoding vector $(Y_1, Y_2, ..., Y_r)$, and $r$ is total number of classes. $Y^u = p(x)$ represents the probability that sample belongs to class $u$. Cross entropy cannot directly

deal with the minimum/compressed representation of $X$ in the hidden layer, because it just considers logits and the true data for statistic computation. Thus, we propose to design a novel loss, which can both evaluate the sufficient representation of input, and evaluate its minimum representation. Its objective is to maximize the mutual information $I(Y, Z_i)$ between input $X$ and true data in the optimization process, and minimize the mutual information $I(Z_{i-1}, Z_i)$ among the hidden layers. Therefore, we can use the Lagrange equation to design the loss based on information bottleneck as:

$$l(w, (X, Y)) = \min_{p(z|x), p(y|z), p(z)} \{I(X; Z) - \beta I(Z; Y)\}, \tag{8}$$

where $\beta$ is the positive Lagrange multiplier to achieve the balance between the complication/compression mutual information of the input representation and the remained relevant information of the network. Other than the sufficient statistic amount and the minimum representation, the decoupling of the hidden factor is another expected feature of the optimal representation, because the independent factor may affect the generation of observing data. Therefore, we aims to decouple the intermediate representation to obtain the finer representation and improve the generalization ability.

In this study, we modify the variational lower bound of information bottleneck by resetting it in the federated learning with randomizing the intermediate representation. In addition, we introduce decoupling in the new loss. The minimization of Lagrange equation in information bottleneck can be maximized equally by the following Lagrange equation as

$$L_{IB}(w) = \max_{p(z|x), p(y|z), p(z)} I(Z; Y, w) - \beta I(Z; X, w), \tag{9}$$

where we aim at learning the fine representation $Z$ of input $X$, which can maximize the representation ability of $Y$ while maximize the compressing ability of $X$. Lagrange parameter makes the optimal balance between relevance and compression. The first term makes $Z$ be predictive of $Y$, and the second term encourages $Z$ to forget $X$. The mutual information $I(Z, Y)$ is calculated as

$$I(Z; Y) = \int dy\, dz\, p(y, z) \log \frac{p(y, z)}{p(y)p(z)} = \int dy\, dz\, p(y, z) \log \frac{p(y|z)}{p(y)}, \tag{10}$$

We design the encoder to define by Markov chain as

$$p(y|z) = \int dx\, p(x, y|z) = \int dx\, p(y|x)\, p(x|z) = \int dx\, \frac{p(y|x)\, p(z|x)\, p(x)}{p(z)}, \tag{11}$$

where we use $q(Y|Z)$ as the variational approximation of $p(Y|Z)$ to solve its incomputability. Since the KL divergence $KL[p(Y|Z), q(Y|Z)] \geq 0$, it can be formulated as

$$\int dy\, p(y|z) \log p(y|z) \geq \int dy\, p(y|z) \log q(y|z). \tag{12}$$

Thus, we can get the expression as

$$\begin{aligned} I(Z; Y) &\geq \int dy\, dz\, p(y, z) \log \frac{q(y|z)}{p(y)} \\ &= \int dy\, dz\, p(y, z) \log q(y|z) - \int dy\, p(y) \log p(y) \\ &= \int dy\, dz\, p(y, z) \log q(y|z) + H(Y). \end{aligned} \tag{13}$$

We can also compute the mutual information $I(Z; X)$ as

$$I(Z; X) = \int dz\, dx\, p(x, z) \log \frac{p(z|x)}{p(z)} = \int dz\, dx\, p(x, z) \log p(z|x) - \int dz\, p(z) \log p(z). \tag{14}$$

We use $k(z)$ as the variational approximation of $p(z)$, because $p(z) = \int dz\, p(z|x)\, p(x)$ maybe have challenges to be calculated. The KL divergence $KL[p(Z), k(Z)] \geq 0$ has the following relationship as

$$\int dz\, p(z) \log p(z) \geq \int dz\, p(z) \log k(z). \tag{15}$$

Thus, we have the following equation as

$$I(Z; X) \leq \int dx\, dz\, p(x)\, p(z|x) \log \frac{p(z|x)}{k(z)}. \tag{16}$$

The loss function can be calculated with the lower bound in practical computation as

$$L \geq \int dx\, dy\, dz\, p(x)\, p(y|x)\, p(z|x) \log q(y|z) - \beta \int dx\, dz\, p(x)\, p(z|x) \log \frac{p(z|x)}{k(z)}. \tag{17}$$

Then we have the following equation as

$$L \approx \frac{1}{M_j} \sum_{i=1}^{M_j} \left[ \int dz\, p(z|x_i) \log q(y_i|z) - \beta\, p(z|x_i) \log \frac{p(z|x_i)}{k(z)} \right], \tag{18}$$

where $p(x, y) = p(x)p(y|x) = \frac{1}{M_j} \sum_{i=1}^{M_j} \theta_{x_i}(x)\theta_{y_i}(y)$, and $\theta_{y_i}(y) = p(y|y_i)$ is the one-hot encoding of the label $y_i$. In the one-hot encoding of the label, zero is replaced by a Gaussian random variable $\xi > 0$, and is normalized to maintain the sum of distribution to be 1. With the reparamierizing technique [42], we have $p(z|x)dz = p(\xi)d\xi$, where $z = f(x, \xi)$ is the deterministic function of $x$ and $\xi$. Then we have the following equation as

$$L_{IB} = \frac{1}{M_j} \sum_{i=1}^{M_j} \mathbb{E}_{\xi \sim p(\xi)} \left[ -\log q(y_i|f(x_i, \xi)) \right] + \beta KL[p(Z|x_i), k(Z)]. \tag{19}$$

For the decoupling of $Z$, we can quantify it with the total relevance measurement $\zeta(Z)$ as

$$\zeta(Z) := KL[p(Z|x_i), \prod_j Q_j(z_j)]. \tag{20}$$

where $\prod_j Q_j(z_j)$ represents the multiplication measurement of $Z$. When $\zeta(Z) = 0$, each component of $Z$ is independent. Thus, we use $\zeta(Z)$ as a regularization term and the minimized equation can be formulated as

$$L_{IB} = \frac{1}{M_j} \sum_{i=1}^{M_j} \mathbb{E}_{\xi \sim p(\xi)} \left[ -\log q(y_i|f(x_i, \xi)) \right] + \beta KL[p(Z|x_i), k(Z)] + \gamma \zeta(Z). \tag{21}$$

By setting $\gamma = \beta$, we can get the following equation as

$$L_{IB} = \frac{1}{M_j} \sum_{i=1}^{M_j} \mathbb{E}_{\xi \sim p(\xi)} \left[ -\log q(y_i|f(x_i, \xi)) \right] + \beta KL[p(Z|x_i), \prod_j Q_j(z_j)]. \tag{22}$$

We set the priori as a factorized Gaussian distribution to naturally or intrinsically obtain the decoupling feature. The first term of the proposed SDFIL quantify the average of the classical cross entropy, and the second term stands for the minimization and decoupling feature of the representation.

## 4 EXPERIMENTS

### 4.1 EXPERIMENTAL SETTING

**Datasets and Models.** To validate the effectiveness of SDFIL, we conducted a series of experiments on the CIFAR10 and CIFAR100 datasets under various conditions of heterogeneity. We utilized VGG9 as the model on the clients, and incorporated batch normalization through time (BNTT) (Kim & Panda, 2021) during network training. Details are in the appendix.

**Gradient Noise.** We added Gaussian noise $\mathcal{N}(0, 1)$ to the gradient, scaled by a noise strength factor $\epsilon$. This factor is gradually increased up to 2, meaning the added noise has a mean of 0 and a standard deviation that reaches 2 at its maximum.

**Implementation Details.** For SDFIL, we have chosen $0.003$ as the value for $\beta$. The SGD optimizer was used with an initial learning rate of $0.1$, reduced by a factor of 5 after $40$, $60$, and $80$ epochs. More details are in the appendix.

**Other Methods.** To demonstrate that our method can effectively enhance the accuracy of federated learning using SNNs, we conducted a comparison under the same settings with the method described in Venkatesha et al. (2021). Furthermore, to prove that our method remains robust and effective in federated learning scenarios characterized by system heterogeneity and gradient noise, we carried out ablation studies. Under identical conditions, the exclusive use of SDFIL algorithm is referred to as w/o SDFIL.

### 4.2 COMPARISON WITH OTHER METHODS

Existing methods (Venkatesha et al., 2021) acknowledge that real-world federated learning systems require strong scalability. They experimented with SNNs and cross-entropy loss functions across a range of client combinations, taking into account the imbalance in data distribution. This study used the proposed SDFIL with the same optimizer and parameter settings. Table 1 shows the final validation accuracy after 100 rounds of training. It is evident that our method outperforms the Venkatesha et al. (2021) under various conditions. In scenarios with full participation of clients, SDFIL only shows a slight advantage. However, as the number of clients increases and the proportion of participating clients remains the same, SDFIL maintains a higher accuracy compared to existing methods, with an improvement of 5% to 10%, demonstrating a significant scalability advantage. Furthermore, under conditions of imbalanced data distribution, SDFIL experiences a smaller drop in accuracy compared to when the data is balanced, indicating robustness.

Then, we analyze the reasons for this phenomenon. SDFIL optimizes the mutual information between the input and output of the model, helping the network filter out task-relevant information while suppressing redundant or irrelevant data. By introducing a decoupled information bottleneck mechanism, SDFIL enhances the model's ability to extract useful information, thereby improving generalization and robustness. Particularly in scenarios with imbalanced client data, SDFIL automatically adjusts to learn the key features from different clients' data, effectively addressing data distribution imbalances and mitigating accuracy degradation. This occurs because irrelevant local noise is suppressed under the constraints of information bottleneck in SDFIL, allowing the model to focus more on global information that contributes to the overall task. Additionally, SDFIL enables the data features of each client to remain relatively independent within the global model, allowing the model to maintain high accuracy even as the number of clients increases. This independence of client features reduces conflicts between different clients' data, contributing to the stability and scalability of the model. By compressing irrelevant information, SDFIL can reduce overfitting to specific clients or local data, thus improving the model's generalization capabilities. By lessening its dependency on particular datasets, the model becomes more robust in the face of noise or imbalanced data, reducing accuracy loss under extreme conditions. Due to the suppression of irrelevant or redundant features, the SDFIL model maintains high stability.

### 4.3 ABLATION STUDY

**Effectiveness on system heterogeneity.** Figure 2 shows the effectiveness of SDFIL on client participation heterogeneity. We only consider the differences in the number of participating clients $P$

Table 1: Final validation accuracy achieved by the models after 100 rounds, for varying numbers of total and participating clients, was evaluated on CIFAR10 and CIFAR100. A key finding is that, as the ratio of total clients to participating clients (K/P) increases, the performance degradation is more pronounced in ANNs and Venkatesha et al. (2021) than in our SDFIL method.

| | Clients (K/P) | | 10/2 | 20/5 | 100/10 | 150/15 | 200/20 |
|---|---|---|---|---|---|---|---|
| CIFAR10 | ANN | IID | 82.81 | 78.25 | 50.84 | 42.82 | 36.37 |
| | (Venkatesha et al., 2021) | non-IID | 79.68 | 73.02 | 44.33 | 36.86 | 33.39 |
| | SNN | IID | 76.44 | 75.01 | 67.54 | 63.85 | 58.76 |
| | (Venkatesha et al., 2021) | non-IID | 73.94 | 68.80 | 58.71 | 59.32 | 55.31 |
| | **SDFIL** | IID | **78.51** | **77.31** | **76.02** | **74.79** | **73.72** |
| | | non-IID | **75.60** | **70.46** | **69.36** | **68.35** | **64.83** |
| CIFAR100 | ANN | IID | 55.56 | 47.47 | 12.29 | 8.25 | 4.61 |
| | (Venkatesha et al., 2021) | non-IID | 53.55 | 44.80 | 13.12 | 8.39 | 5.08 |
| | SNN | IID | 47.25 | 49.95 | 42.79 | 36.61 | 32.52 |
| | (Venkatesha et al., 2021) | non-IID | 41.00 | 46.64 | 40.91 | 37.35 | 32.13 |
| | **SDFIL** | IID | **50.61** | **51.06** | **48.06** | **46.22** | **43.60** |
| | | non-IID | **44.13** | **47.11** | **45.54** | **44.68** | **43.40** |

and a type of data imbalance ($\alpha = 0.5$), without taking gradient noise into account ($\epsilon = 0$). In the IID scenario, since the data is evenly distributed across clients, adding SDFIL results in only a slight increase in accuracy. However, for the non-IID scenario, our method still retains a certain degree of generalization capability even with fewer participants, particularly under extreme conditions such as an $K/P$ ratio of 100/1, where the accuracy can be improved by up to 5.21%.

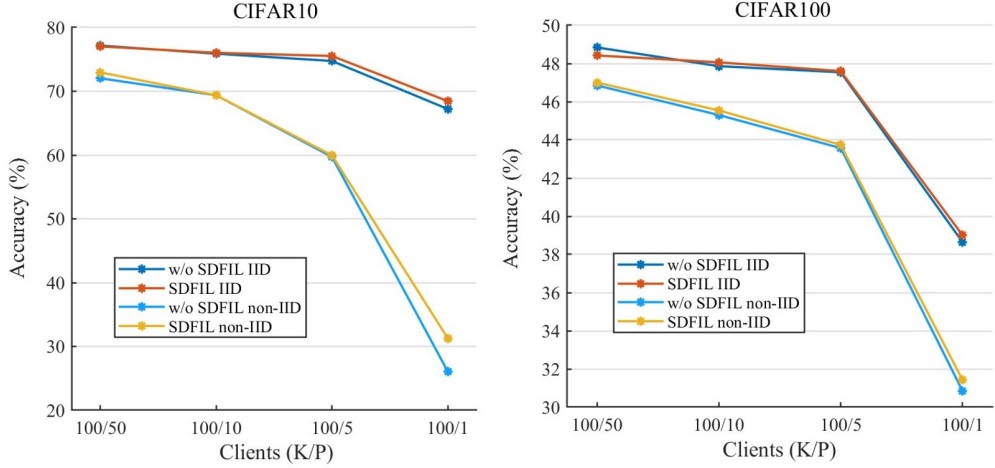

Figure 2: Effectiveness on client participation heterogeneity. A key finding is that, SDFIL still retains a certain degree of generalization capability even under extreme conditions such as an $K/P$ ratio of 100/1.

Furthermore, as shown in Figure 3, we also do not introduce gradient noise here. Under extremely unbalanced non-IID conditions ($\alpha = 0.25$), although the reduction in accuracy due to the decrease in the number of participating clients from $P = 10$ to $P = 5$ is inevitable, our method minimizes the relative decrease in accuracy. This indicates that under conditions of significant data heterogeneity and client participation heterogeneity, the gradients produced by SDFIL enhance the robustness of the trained model. Moreover, when the data distribution is relatively balanced ($\alpha = 1.0, 2.0, 4.0$), SDFIL not only improves accuracy but also shows less susceptibility to changes of data distribution,

resulting in a more stable relative decrease in accuracy. Specifically, while the accuracy is increased by up to 1%, it consistently achieves the relative decrease of approximately 2.5%.

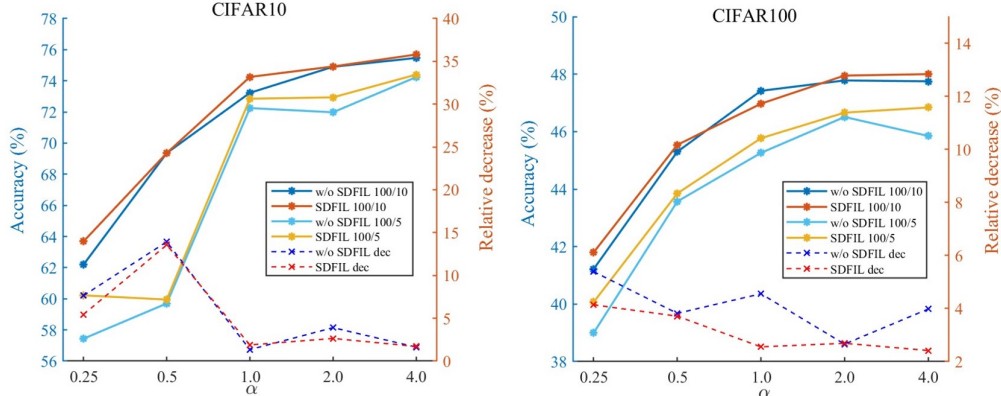

Figure 3: Effectiveness on data heterogeneity. 'dec' indicates the relative decrease in accuracy from $100/10$ to $100/5$. A key finding is that, SDFIL not only performs better but also minimizes the relative decrease in accuracy even under conditions of significant data heterogeneity and client participation heterogeneity.

**Effectiveness on gradient noise.** We introduce gradient noise at the end, while simultaneously considering both types of heterogeneity. Table 2 and Table 3 displays the effects of incorporating gradient noise under conditions of data heterogeneity and client participation heterogeneity on CIFAR10 and CIFAR100. The more imbalanced the data distribution among clients, the greater the impact of noise on the model's performance. Under the same conditions, SDFIL shows an improvement in accuracy by up to 5.14%, indicating that the gradients produced by the improved loss function are better suited to the addition of random noise. This is because SDFIL extracts task-relevant latent features from the data while suppressing or discarding task-irrelevant noise or redundant information. Through SDFIL, the model selectively captures the most useful feature representations, thereby effectively reducing the impact of noise on the global model. This makes the global model more robust to noise when aggregating gradients and better able to adapt to diverse heterogeneous data distributions. This allows the use of noisy gradients to effectively implement privacy protection.

Table 2: Final validation accuracy achieved by the models after 100 rounds on CIFAR10.

| Clients (K/P) | 100/10 | | | | 100/5 | | | |
|---|---|---|---|---|---|---|---|---|
| Methods | w/o SDFIL | | **SDFIL** | | w/o SDFIL | | **SDFIL** | |
| $\alpha$ | $\epsilon=1$ | $\epsilon=2$ | $\epsilon=1$ | $\epsilon=2$ | $\epsilon=1$ | $\epsilon=2$ | $\epsilon=1$ | $\epsilon=2$ |
| 0.25 | 62.37 | 64.05 | **64.57** | **65.00** | 57.32 | 60.08 | **62.46** | 58.61 |
| 0.5 | 69.51 | 68.76 | **71.04** | **69.88** | 61.94 | 56.77 | **66.57** | **59.35** |
| 1.0 | 73.24 | 73.01 | **73.68** | **73.38** | 72.34 | 72.59 | **72.49** | **73.01** |
| 2.0 | 74.57 | 74.71 | **74.72** | **75.18** | 73.02 | 72.11 | **73.27** | **72.30** |
| 4.0 | 75.31 | 75.30 | **75.58** | **75.80** | 75.30 | 74.45 | **75.46** | **74.74** |

## 4.4 ENERGY CONSUMPTION

We approximate the energy use of 32-bit integer arithmetic in a 45nm CMOS process, focusing only on Multiply and Accumulate (MAC) operations and excluding memory and peripheral circuit energy, as detailed in Horowitz (2014). Specific details are included in the appendix. Figure 4 displays the estimated energy consumption for each layer in ANN and SDFIL using the VGG9

Table 3: Final validation accuracy achieved by the models after 100 rounds on CIFAR100.

| Clients (K/P) | 100/10 | | | | 100/5 | | | |
|---|---|---|---|---|---|---|---|---|
| Methods | w/o SDFIL | | **SDFIL** | | w/o SDFIL | | **SDFIL** | |
| $\alpha$ | $\epsilon = 1$ | $\epsilon = 2$ | $\epsilon = 1$ | $\epsilon = 2$ | $\epsilon = 1$ | $\epsilon = 2$ | $\epsilon = 1$ | $\epsilon = 2$ |
| 0.25 | 42.35 | 43.09 | **43.77** | **43.29** | 40.00 | 40.75 | **40.51** | **40.93** |
| 0.5 | 45.71 | 45.55 | **45.93** | **46.67** | 43.92 | 43.64 | **44.06** | **44.12** |
| 1.0 | 46.58 | 47.09 | **46.92** | **47.73** | 45.55 | 45.07 | **45.60** | **45.85** |
| 2.0 | 47.47 | **47.71** | **47.92** | 47.65 | 45.59 | 46.13 | **47.01** | **46.97** |
| 4.0 | 48.05 | 48.42 | **48.84** | **49.80** | 46.32 | 46.58 | **46.87** | **47.16** |

model trained on CIFAR10 for 100/10 clients with non-IID data distribution. The total energy estimated by ANN is approximately $227.99\mu J$, in contrast to SDFIL, which is estimated at $29.76\mu J$, making it 7.7 times more energy-efficient. Unlike the constant energy consumption of the ANN across all instances, the energy usage of the SNN fluctuates with each instance, influenced by the activity of its spikes.

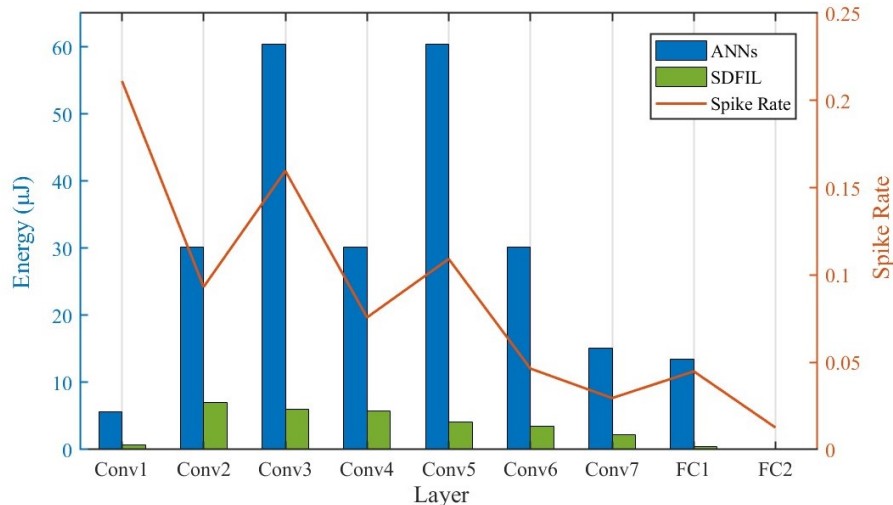

Figure 4: Comparison of energy consumption. A key finding is that, SDFIL are considerably more energy efficient compared to ANNs.

## 5 CONCLUSION

This study introduces and evaluates a approach to robust, spike-based federated learning within environments characterized by noise, data heterogeneity, and client participation heterogeneity. The proposed SDFIL not only maximizes mutual information between ground truths and model predictions but also minimizes it between intermediate representations. Experimental results demonstrate that the SDFIL framework outperforms existing methods in terms of robustness to noise across various settings and degrees of heterogeneity. By mitigating the adverse effects of noisy and inconsistent data, SDFIL significantly enhances the feasibility and energy efficiency of federated learning on energy-constrained devices. These findings underscore the promising applications of SNNs in federated learning scenarios. Given the promising effectiveness of SDFIL, it also has limitations, including lack of investigation on different SNN architectures, other types of noises, and tests on real-world devices. Limitations and future work are discussed in detail in Appendix A.5.

## ETHICS STATEMENT

This study explores the integration of SNNs with federated learning to enhance image recognition tasks using the publicly available CIFAR10 and CIFAR100 datasets. These datasets consist of non-sensitive, public domain images widely used in the computer vision community, complying with all applicable terms of use. We have taken careful measures to adhere to data protection and privacy regulations, including GDPR, despite the non-sensitive nature of the data. The federated learning approach further bolsters our commitment to privacy as it allows for collaborative learning without direct data sharing. Our research team is committed to upholding the highest standards of research integrity and ensuring that our findings accurately reflect the conducted experiments and analyses.

## REPRODUCIBILITY

We provide code for all experiments with the submission and provide the required experimental details in the appendix.

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

# A  APPENDIX

## A.1  TRAINING SPIKING NEURAL NETWORKS WITH BNTT

In reference Kim & Panda (2021), the authors developed a strategy that applies batch normalization along the time dimension to boost the training of SNNs. This technique enables the creation of high-performance, low-latency SNNs from scratch, without relying on previously trained ANN models. The key to this method is the introduction of a distinct learning parameter for each time-step, which expands the batch norm layer through time. During the forward propagation, the BNTT is applied after each layer as:

$$u_i^t = \lambda u_i^{t-1} + \text{BNTT}_{\gamma^t}\left(\sum_{j \in N} w_{ij} o_j^t\right) = \lambda u_i^{t-1} + \gamma_i^t \left(\frac{\sum_{j \in N} w_{ij} o_j^t - \mu_i^t}{\sqrt{(\sigma_i^t)^2 + \epsilon}}\right) \tag{A1}$$

where $\mu_i^t$ and $\sigma_i^t$ represent the mean and variance of the mini-batch $\mathbb{B}$ at time step t. These are computed using an exponential moving average across the training phases, which are applied to standardize the validation set during inference. The scaling factor $\gamma$ is adjusted through backpropagation, with distinct $\gamma^t$ values assigned at each time step for optimized inference.

## A.2  INTRODUCTION AND PROCESSING OF DATASETS

**CIFAR10** Introduced by Krizhevsky (2009), the CIFAR10 dataset comprises a collection of 60,000 color images categorized into 10 distinct classes, each containing 6,000 images. This dataset is segmented into two subsets: 50,000 images for training and 10,000 for testing, distributing 5,000 training images and 1,000 testing images per class. All images are presented in a uniform resolution of 32x32 pixels.

**CIFAR100** The CIFAR100 dataset expands upon the CIFAR-10 dataset, offering a more complex and varied testing ground for image recognition systems. It features 100 distinct classes, significantly broadening the scope beyond the 10 classes found in CIFAR10. This variety includes a diverse array of objects and concepts, providing a rigorous challenge in image classification.

**Poisson Rate Coding** The training process starts by encoding the pixel values into spike trains of length $T$ using Poisson rate coding. In Poisson coding, the relationship between the number of spikes and the pixel intensity is established. At each time step, a random value is sampled from the minimum to maximum possible pixel intensity range $I_{min}$ to $I_{max}$. If this sampled value is below the actual pixel intensity, a spike is generated, rendering the spike occurrence at each time step stochastic. The accumulated number of spikes over time steps proportionally represents the pixel intensity. Thus, when spikes are aggregated over time, the resulting image closely approximates the original.

## A.3 DETAILED EXPERIMENTAL SETUP AND RESULTS

**Detailed Setup** For client participation heterogeneity, we set up the case trained with 100 clients and gradually decrease the number of participating clients from 50 to 1. Specifically, we employ two $K/P$ scenarios, 100/10 and 100/5, to combine with other issues. To address data heterogeneity, we employed a Dirichlet distribution to generate non-IID data, which was then distributed among various clients. We initiate with $\alpha = 4$ and reduce it incrementally by half, ensuring the model's stability, to assess the system's performance. The settings of the hyper-parameters in the experiment are as Table 4.

Table 4: Settings of the hyper-parameters.

| Description | Value |
|---|---|
| Rounds of training | 100 |
| Number of local epochs | 5 |
| Training batch size | 32 |
| Testing batch size | 64 |
| Initial learning rate | 0.1 |
| Reduction factor for learning rate | 5 |
| Weight decay for SGD | 5e-4 |
| SGD momentum | 0.95 |
| Timesteps | 20 |
| kernel size | 3 |

**Detailed Results** For the effectiveness of SDFIL on system and data heterogeneity, the detailed results are as Table 5 and Table 6.

Table 5: Detailed results for the effectiveness of SDFIL on client participation heterogeneity.

| Clients (K/P) | | | 100/50 | 100/10 | 100/5 | 100/1 |
|---|---|---|---|---|---|---|
| CIFAR10 | w/o SDFIL | IID | **77.15** | 75.88 | 74.74 | 67.21 |
| | | non-IID | 72.03 | 69.36 | 59.70 | 26.07 |
| | **SDFIL** | IID | 77.02 | **76.02** | **75.52** | **68.44** |
| | | non-IID | **72.95** | 69.36 | **59.95** | **31.28** |
| CIFAR100 | w/o SDFIL | IID | **48.84** | 47.86 | 47.54 | 38.65 |
| | | non-IID | 46.85 | 45.30 | 43.57 | 30.86 |
| | **SDFIL** | IID | 48.42 | **48.06** | **47.60** | **39.01** |
| | | non-IID | **47.00** | **45.54** | **43.74** | **31.43** |

Table 6: Detailed results for the effectiveness of SDFIL on data heterogeneity.

| | | $\alpha$ | 0.25 | 0.5 | 1.0 | 2.0 | 4.0 |
|---|---|---|---|---|---|---|---|
| CIFAR10 | w/o SDFIL | 100/10 | 62.20 | 69.36 | 73.23 | 74.90 | 75.47 |
| | | 100/5 | 57.44 | 59.70 | 72.25 | 71.98 | 74.24 |
| | **SDFIL** | 100/10 | **63.69** | **69.36** | **74.25** | **74.92** | **75.69** |
| | | 100/5 | **60.22** | **59.95** | **72.85** | **72.94** | **74.40** |
| CIFAR100 | w/o SDFIL | 100/10 | 41.22 | 45.30 | **47.42** | 47.78 | 47.75 |
| | | 100/5 | 39.00 | 43.57 | 45.26 | 46.51 | 45.85 |
| | **SDFIL** | 100/10 | **41.80** | **45.54** | 46.97 | **47.95** | **48.00** |
| | | 100/5 | **40.07** | **43.85** | **45.77** | **46.66** | **46.84** |

## A.4 ESTIMATED ENERGY

This approximation is relatively coarse since it solely accounts for Multiply and Accumulate (MAC) operations, disregarding the energy consumption of memory and peripheral circuits. The energy costs associated with these operations are detailed in Table 7. For a convolutional layer with kernel size $k \times k$, $I$ input channels, and $O$ output channels, operating on an $N \times N$ input feature map and producing an $M \times M$ output feature map, the total number of operations (OPS) is expressed as:

$$\text{OPS} = M^2 \times I \times k^2 \times O \tag{A2}$$

For a fully connected layer with $I$ inputs and $O$ outputs, the number of operations (OPS) is:

$$\text{OPS} = I \times O \tag{A3}$$

Given that SNNs utilize binary spikes, the MAC operations simplify to accumulation (AC) operations, thereby enhancing energy efficiency. The energy consumption for ANNs can be directly calculated as:

$$E_{\text{ANN}} = \text{OPS} \times E_{\text{MAC}} \tag{A4}$$

The energy usage for SNNs is calculated by multiplying the OPS by the spiking rate $R$ across all timesteps $T$, formulated as:

$$E_{\text{SNN}} = \text{OPS} \times R \times T \times E_{\text{AC}} \tag{A5}$$

Table 7: Energy estimation for multiply and accumulate operations.

| Operation | Estimated Energy (pJ) |
|---|---|
| 32-bit Multiply $E_{\text{Mult}}$ | 3.1 |
| 32-bit Add $E_{\text{Add}}$ | 0.1 |
| 32-bit Multiply and Accumulate $E_{\text{MAC}}$ | 3.2 |
| 32-bit Accumulate $E_{\text{AC}}$ | 0.1 |

Table 8 shows the spike rate of the corresponding layers for the final model of SDFIL after 100 rounds.

Table 8: Spike rate of the corresponding layers.

| Layers | Conv1 | Conv2 | Conv3 | Conv4 | Conv5 | Conv6 | Conv7 | FC1 | FC2 |
|---|---|---|---|---|---|---|---|---|---|
| Spike rate | 0.2112 | 0.0932 | 0.1597 | 0.0757 | 0.1092 | 0.0464 | 0.0295 | 0.0448 | 0.0126 |

### A.5 LIMITATIONS AND FUTURE WORK

Given the promising results demonstrating the effectiveness of SDFIL, it also has limitations.

First, the effectiveness of SDFIL may be highly dependent on the specific SNN architecture and hyperparameter settings. Different network structures may exhibit varying adaptability under non-independent and identically distributed (non-IID) data and client heterogeneity conditions. The current research may not have fully explored the diversity of SNN architectures. Future studies could attempt to validate the effectiveness of SDFIL across a wider range of SNN architectures, investigating how different architectures impact the disentanglement results.

Second, while SDFIL demonstrates strong robustness in handling gradient noise, its performance may degrade in extreme noise environments (such as very high noise levels or adversarial attacks). Different types of noise, such as asynchronous client updates or data contamination introduced by malicious clients, may require further research and optimization.

Finally, the current experiments are primarily based on simulated environments, and the framework has not yet been thoroughly tested in large-scale real-world federated learning systems. In particular, energy consumption evaluations may lack rigorous validation on real resource-constrained devices. Future work could further test SDFIL in real IoT or mobile device scenarios to verify its performance under actual energy consumption and computational resource constraints.

Future work can include two aspects.

First, the current research primarily focuses on optimizing single-task federated learning. Future work could consider extending SDFIL to multi-task federated learning environments, where important features can be shared and disentangled across different tasks. This would help improve the model's generalization ability in multi-task and cross-domain applications.

Second, future research could explore the application of SDFIL in more complex federated learning scenarios, such as asynchronous update mechanisms, dynamic client participation, and data sparsity. These situations are common in real-world applications, and enhancing the algorithm's adaptability and generalization in these contexts will be an important research direction.

### A.6 THEORETICAL ANALYSIS

**Settings:** In this section, we analyze the theoretical convergence results of the designed loss function and the robustness of the aggregation scheme For loss convergence analysis, we rewrite the entire loss function $L_{IB}$ as follows:

$$l_{IB} = CE_{\text{loss}} + \beta KL_{\text{loss}} \tag{A6}$$

where $CE_{\text{loss}}$ is the sufficient statistic loss component, and $KL_{\text{loss}}$ is the decoupled compressive statistic loss component. The LIF model simulates the accumulation of a neuron's membrane potential in response to input currents, firing a spike when a threshold is reached. First, we derive the optimal cost for each loss component and demonstrate the corresponding convergence rate. Then, we compare the optimal costs of each component to analyze the convergence behavior of the entire loss function.

**Sufficient Statistic Loss**

**Proposition 1**: The sufficient statistic loss ($CE_{\text{loss}}$) in the SDFIL loss function is equivalent to the classical cross-entropy loss, whose optimal convergence rate is up to the logarithmic factor.

**Proof**: Let a single sample-label pair be $(x, y)$. In the LIF model, we can calculate the optimal cost of $CE_{\text{loss}}$ associated with the $J_{IB}$ loss function for the sample as follows:

$$CE_{\text{loss}} = \mathbb{E}_{\xi \sim p(\xi)} \left[ -\log q \left( y \mid f(x, \xi) \right) \right] \tag{A7}$$

In the LIF model, the activation function simulates the accumulation of the membrane potential and the spiking process, with the dynamics of the membrane potential described by differential equations. Let $Y$ be the one-hot encoded vector, $r$ the total number of classes, and $\hat{Y}_u = f(x, \xi)$ the probability that the sample is in class $u$. Then, Eq. (24) can be rewritten as:

$$CE_{\text{loss}} = -\sum_{u=1}^{r} Y_u \log(\hat{Y}_u) \tag{A8}$$

Clearly, this equation is identical to Eq. (7), which is the classical cross-entropy loss calculation. Therefore, $CE_{\text{loss}}$ in our designed loss function is completely equivalent to the traditional cross-entropy loss in deep learning and federated learning, and it achieves the same optimal convergence rate, up to a logarithmic factor. This proves Proposition 1.

**Decoupled Compressive Statistic Loss**

We now calculate the optimal cost of the decoupled compressive statistic loss $KL_{\text{loss}}$ associated with the SDFIL loss function, considering the LIF spiking neuron model as the activation function. The LIF model simulates the accumulation and leakage of the membrane potential during the spiking process. Due to the dynamic characteristics of the membrane potential in the LIF model, we introduce a time-dependent prior distribution to describe the noise accumulation in the membrane potential. We recommend adopting a noise distribution, such as a log-normal distribution related to the spike rate, as the prior $Q(z)$, which better captures the firing characteristics of neurons.

**Proposition 2:** Let $z = f(x, \xi) = \xi \cdot f(x)$, where $\xi \sim p_\alpha(\xi)$, and $p_\alpha$ is the parametric noise distribution. For the LIF activation function, let the probability distribution of $z$ be $p(z) = Q_0\delta_0(z) + \frac{C_{\text{nst}}}{z}$. Now, if $f(x) \neq 0$, then:

(i)
$$KL_{\text{loss}} = -H(p_\alpha(x)(\log \xi)) + \log(C_{\text{nst}}) \tag{A9}$$

(ii)
$$KL_{\text{loss}} = -\log \alpha_\theta(x) + \text{const} \tag{A10}$$
while $p_\alpha(\xi) = \log \mathcal{N}(0, \alpha_\theta^2(x))$ follows a log-normal distribution; otherwise,

(iii)
$$KL_{\text{loss}} = -\log Q. \tag{A11}$$

For the LIF activation function, which models spiking neurons based on accumulated membrane potential dynamics, $p_\alpha(\xi)$ describes the distribution of the noise involved in triggering a spike (i.e., reaching the threshold). If the LIF model involves the log-normal distribution of noise, similar calculations as used for ReLU and Softplus can be extended to LIF as well, but with the membrane potential dynamics in mind.

**Proof:** We use the following more generic KL divergence definition, since $P_\theta(z|x)$ and $P_\theta(z)$ are not absolutely continuous:

$$KL[P(z), \hat{Q}(z)] = \int \log \frac{dP}{d\hat{Q}} dP \tag{A12}$$

where $P \ll \hat{Q}$. Since KL divergence is invariant under invertible parameter transformations, we can express this using a typical invertible transformation $\psi(z)$:

$$KL[P(z), \hat{Q}(z)] = KL[P(\psi(z)), \hat{Q}(\psi(z))] \tag{A13}$$

Now, assume $f(x) \neq 0$, then $z \neq 0$. Since $P(\log z) = C_{\text{nst}}$ when $z > 0$, we get:

$$KL_{\text{loss}} = KL[P_\theta(z|x), P_\theta(z)] = KL[P_\theta(\log z|x), P_\theta(\log z)] \tag{A14}$$

This proves Proposition 2(i). Now, when $p_\alpha(z)(\xi) = \log \mathcal{N}(0, \alpha_\theta^2(x))$, we obtain $p_\alpha(x)(\log \xi) = \mathcal{N}(0, \alpha_\theta^2(x))$, and the entropy of the Gaussian distribution is given by:

$$H(\mathcal{N}(0, \alpha)) = \log \alpha_\theta(x) + \frac{1}{2} \log(2\pi e) \tag{A15}$$

This proves Proposition 2(ii).

In the case of the LIF model, the dynamics of membrane potential (which includes both integration and leakage) must be factored into the overall model. Hence, the noise required to trigger a spike

plays a role in shaping the distribution, analogous to the role that activation functions like ReLU and Softplus play in traditional neural networks.

Next, if we set $f(x) = 0$, then $z = 0$, so $p(z|x) = \delta_0(z)$. After that, we can compute:

$$KL_{\text{loss}} = \int \log \frac{dP(z|x)}{dP(z)} dP = -\log Q \tag{A16}$$

Similarly, in the LIF model, we can compute the optimal convergence rate of $KL_{\text{loss}}$. This proves Proposition 2(iii).

**SDFIL Loss Case**

**Proposition 3**: The optimal convergence rate of the SDFIL loss function using the LIF activation is up to the logarithmic factor.

**Proof**: We can write the collated optimal cost of the SDFIL loss function ($L_{IB}$) for a single sample $(x_i, y_i)$ using the LIF activation model as follows. In the LIF model, neurons accumulate membrane potential over time and emit spikes once the potential reaches a certain threshold, after which the membrane potential resets. The leak dynamics of the membrane potential are modeled by an exponential decay process, which adds complexity to the loss function in comparison to ReLU or Softplus.

Using the LIF model, the loss function is modified to account for the dynamic behavior of the neuron membrane potential. For LIF neurons, the collated optimal cost of $L_{IB}$ can be expressed as:

$$L_{IB} = \frac{1}{M_j} \sum_{i=1}^{M_j} \mathbb{E}_{\xi \sim p(\xi)} \left[ -\log q(y_i \mid f(x_i, \xi)) \right] + \beta \log \alpha_\theta(x_i) \tag{A17}$$

where $\alpha_\theta(x_i)$ is associated with the membrane potential's leakage and spike threshold dynamics. The exponential decay in membrane potential and stochastic behavior related to spiking are taken into account here.

While the LIF activation inherently has more complex dynamics due to the temporal evolution of the membrane potential and its eventual spike, the loss function maintains a similar structure to that used in ReLU activations but incorporates the time constants related to the membrane potential's leak rate.

In the LIF model, the collated optimal cost of $L_{IB}$ becomes:

$$L_{IB} = \frac{1}{M_j} \sum_{i=1}^{M_j} \mathbb{E}_{\xi \sim p(\xi)} \left[ -\log q(y_i \mid f(x_i, \xi)) \right] + \beta \left[ \frac{1}{2\sigma^2} \left( \sigma^2(x_i) + \mu^2 \right) - \log \frac{\alpha(x_i)}{\sigma} - \frac{1}{2} \right] \tag{A18}$$

Here, $\alpha(x_i)$ models the stochastic behavior of the LIF neuron related to the timing of spikes and the leak rate of the membrane potential. This form reflects the time-varying characteristics of the LIF neurons and their impact on the optimal loss function.

From the above theoretical results, it is demonstrated that the optimal convergence rate for $L_{IB}$ is up to the logarithmic factor, based on the proofs of Proposition 1 and Proposition 2. This proves Proposition 3 and, therefore, we can conclude that our method converges under the LIF model.

