# OpenReview forum: "Robust Spike-based Decoupled Federated Information Bottleneck Learning with Spiking Neural Network under System Heterogeneity"
_ICLR.cc/2025/Conference — ICLR 2025 Conference Withdrawn Submission_

### Official Review · Reviewer_2GJz · 2024-10-18

**Soundness:** 2
**Presentation:** 2
**Contribution:** 1
**Rating:** 3
**Confidence:** 4

**Summary:**

This paper introduces a new federated learning framework called SDFIL, aimed at improving the performance and robustness of Spiking Neural Networks (SNNs) in heterogeneous systems. The paper aims to address challenges in federated learning: client participation heterogeneity, data heterogeneity, and gradient noise. By incorporating the information bottleneck principle, the framework reduces redundant information in intermediate layers, enhancing the model's resistance to noise and outliers.

The authors validate the effectiveness of this approach through experiments on CIFAR10 and CIFAR100 datasets under various noise and heterogeneity conditions Experimental results show that SDFIL improves accuracy by 5%-10% and boosts energy efficiency by up to 7.7 times compared to traditional methods.

**Strengths:**

1.The combination of federated learning and SNN has practical significance. Such a combination is expected to fully utilize distributed data training models on low-energy embedded devices.
2.The authors demonstrated the effectiveness of the SDFIL framework through small-scale experiments. Compared with other SNN federated learning methods, the accuracy and energy efficiency on CIFAR10 and CIFAR100 are improved.

**Weaknesses:**

1.From the perspective of SNN: The relationship between the core method proposed in this paper and SDFIL is unclear. There is no part in eq 6-22 that is intrinsically related to SNN. In my opinion, the SNN mentioned in this paper is only loosely coupled, and at most proves the feasibility of combining this framework with SNN. The low energy consumption advantage demonstrated by the authors in 4.4 seems to benefit from SNN and has little to do with the proposed method.

2.From the perspective of Theory: The core contribution of the paper, ie then eq 6-22, appears rather trivial. The application of the Information Bottleneck (IB) framework to SNNs and federated learning follows standard techniques. The core ideas of maximizing mutual information and minimizing redundant representations are well-established, and the paper fails to convincingly justify the need for these extensions specifically in the context of SNNs. The theoretical derivation is largely similar to traditional variational lower bounds and does not provide substantial innovation targeting the unique properties of SNNs.

3.From the claim of the author: "we methodically address client participation heterogeneity, data heterogeneity, and gradient noise". The introduction of "decoupling" in eq 20-22 is not sufficiently justified. While decoupling hidden representations might intuitively seem beneficial, the theoretical basis for its effectiveness within SNNs and federated learning is weak. The authors provide no rigorous analysis demonstrating how or why this approach would enhance robustness to noise or improve model scalability under system heterogeneity, making the proposed method appear more like a direct adaptation of existing IB models.

4.From the experimental perspective: The experiments on cifar10 and cigar100 are too small to fully demonstrate the advantages and necessity of federated learning as claimed by the authors. Please consider at least adding larger-scale experiments such as imagenet. Although in the context of federated learning, the method proposed by the authors shows advantages, the acc of ~75% on cifar10 and ~50% on cifar100 make it difficult to believe that the algorithm has the ability to scale to larger real-world datasets.

**Questions:**

See weakness.

---

### Official Review · Reviewer_hAfr · 2024-10-18

**Soundness:** 1
**Presentation:** 1
**Contribution:** 2
**Rating:** 1
**Confidence:** 4

**Summary:**

The paper proposes a federated learning framework SDFIL for SNNs.

**Strengths:**

The paper uses the classical information bottleneck loss to train an SNN with federated learning.

**Weaknesses:**

W1. The method section (Section 3) is almost identical to the approach presented in [1], yet this paper does not cite [1]. Since [1] was published in 2017, the novelty of this paper is quite limited. SDFIL merely applies the method from [1] to SNNs, without introducing any specific design tailored for SNNs.
>[1] Alexander A. et al. Deep Variational Information Bottleneck. ICLR 2017.

W2. The presentation of this paper is lacking, suggesting that it hasn't been carefully prepared. Numerous typos significantly hinder the readability. Only a few examples are listed here.
>W2-(1): Line 185-186. ‘Here, $\alpha$ ...’ is mentioned, but there is no prior reference to $\alpha$ in the text.

>W2-(2): Line 208. '$F_j$' should be $M_j$.

>W2-(3): Line 219-220, and Eq. (8). The text refers to $Z_{i-1},Z_{i}$ but Eq. (8) does not include $Z_i$. Additionally, the phrase '$I(Y,Z_i)$ between input $X$ and true data' appears to be an incorrect description.

>W2-(4): Line 240-322. The use of symbols are confusing, as the pairs $x-X$, $y-Y$, $z-Z$ are used interchangeably. In most cases, the text uses capital letters, while the equations use lowercase ones.

>W2-(5): Line 293 mentioned 'where $p(x,y)$....', but there is no $p(x,y)$ in Eq. (18).

**Questions:**

Q1. How does the mutual information-based loss function "comprise the sufficiency of the information bottleneck, compressed statistical amount, and decoupling feature in federated learning of the local model," as claimed in Section 3.2?

Q2. Apart from reference [2] mentioned in the paper, several other relevant works [3-5] are missing.
>[2] Venkatesha et al. Federated learning with spiking neural networks. IEEE Transactions on Signal Processing, 2021.

>[3] Xie et al. Efficient Federated Learning With Spike Neural Networks for Traffic Sign Recognition.  IEEE Transactions on Parallel and Distributed Systems, 2022.

>[4] Skatchkovsky et al. Federated Neuromorphic Learning of Spiking Neural Networks for Low-Power Edge Intelligence. ICASSP, 2020.

>[5] Liu et al. Federal SNN Distillation: A Low-Communication-Cost Federated Learning Framework for Spiking Neural Networks. Journal of Physics: Conference Series, 2022.

---

### Official Review · Reviewer_J5nh · 2024-10-29

**Soundness:** 3
**Presentation:** 3
**Contribution:** 3
**Rating:** 6
**Confidence:** 5

**Summary:**

This paper introduces a novel framework, spike-based decoupled federated information-bottleneck learning (SDFIL), designed to improve robustness and energy efficiency in federated learning environments utilizing spiking neural networks (SNNs) under conditions of system heterogeneity. The authors propose a decoupled information bottleneck approach to address issues arising from non-IID data and variable client participation, demonstrating improvements in model accuracy and power consumption.

**Strengths:**

1. The integration of SNNs with federated learning to address system heterogeneity represents a significant contribution to enhancing the scalability and efficiency of distributed machine learning systems.
2. The proposed SDFIL framework is well-founded, with a clear explanation of the decoupled information bottleneck technique that efficiently handles outliers in data.
3. Extensive experiments conducted on CIFAR-10 and CIFAR-100 datasets demonstrate the effectiveness of SDFIL over existing methods, particularly in scenarios of high noise and data heterogeneity.
4. The study provides a compelling analysis of energy consumption, showing SDFIL's potential to significantly reduce power usage, which is critical for deployment on edge devices.

**Weaknesses:**

1. There is insufficient discussion regarding the sensitivity of the method to the hyperparameter of loss function and the impact of parameter tuning on performance. This aspect is critical for understanding how the model behaves under various parameter settings.
2. The explanation of the assumptions behind the model's design and discussions about the conditions that might restrict its applicability are quite brief. A more detailed exploration in the methods section would help clarify these limitations.
3. It would be beneficial if the authors could provide additional resources that detail the architecture of the model and the data processing techniques used. This would aid other researchers and practitioners in better understanding and applying the SDFIL model.

**Questions:**

1. Could the authors explain how variations in the loss function’s hyperparameter impact the model's performance? Are there particular parameter settings that are crucial for the model's effectiveness?
2. Could the authors provide more information on the limitations imposed by the model's assumptions?
3. Would it be possible for the authors to share more detailed descriptions of the model's architecture and the steps involved in processing the datasets?

---

### Official Review · Reviewer_NwZZ · 2024-11-01

**Soundness:** 2
**Presentation:** 2
**Contribution:** 2
**Rating:** 3
**Confidence:** 4

**Summary:**

This paper combines federated learning and SNNs, and proposes a spike-based federated learning method. Results are validated on CIFAR-10 and CIFAR-100 in several settings.

My key concern is the motivation of the paper: The SNNs on embedded devices are strongly emphasized in the paper, but two cited papers are not about this topic and did not mention "embedded device" even once; federated learning and federated learning in SNNs are introduced without any reasons on why they are necessary to be introduced. The author just jumps to these concepts suddenly, even if the previous paragraph/sentence is about another different concept.

**Strengths:**

1. The proposed method SDFIL is introduced in detail with many equations.

2. The method is validated on several datasets and settings

**Weaknesses:**

1. Line 52 "Due to this power efficiency, SNNs show great potential for deployment in
embedded devices (Kucik & Meoni, 2021; Ottati et al., 2023). However, embedded devices often
require continuous data collection for training and updates.". I checked these two papers, but none of them are about SNNs on embedding devices. As "embedded device" is a keyword in the abstract and introduction. Please double-check it and either provide more relevant citations that directly discuss SNNs in embedded devices, or explain how to draw this conclusion if it's not explicitly stated in the cited papers.

2. Line 17 "In federated learning scenarios, where multiple energy-constrained devices
collaborate, adopting efficient SNN models with effective training methods is critical. " The story presented in the abstract is not clear. Why the federated learning introduced here without any explanation of its necessity? I suggest that the authors provide a clearer motivation for why federated learning is necessary in this context, perhaps by explaining the specific challenges or limitations that federated learning addresses in SNN applications.

3. Line 63 "When applying federated learning to SNNs, three key challenges arise, as shown in Figure 1. First,
data heterogeneity among clients is particularly problematic for SNNs". The same problem as the above point: why it is necessary to apply federated learning to SNNs. Please provide a clear explanation of the specific benefits or necessities of applying federated learning to SNNs before discussing the challenges. This would help readers understand the motivation behind combining these two concepts.

**Questions:**

See Weakness for questions

---

### Official Review · Reviewer_BwDy · 2024-11-03

**Soundness:** 2
**Presentation:** 2
**Contribution:** 1
**Rating:** 3
**Confidence:** 4

**Summary:**

The paper propose a federated learning framework, spike-based decoupled federated information-bottleneck learning (SDFIL), to enable robust, low-power federated learning through SNNs under system heterogeneity. The paper conduct experiments including the federated
learning  performance comparison, ablation study, effectiveness on gradient noise, energy consumption.

**Strengths:**

The paper introduce the information bottleneck to improve the performance of SNN for federated learning. And the experiment part is comprehensive, including the non-IID setting and many other aspects.

**Weaknesses:**

1.The novelty of the paper is limited and the paper lacks essential references. This method is just a combination of [1] and [2].  Most formulas of the method in section 3.2 are all borrowed from paper[1], but not referenced.

2.About the experiment part, there are only two methods for comparison in Table 1. More experiments of other kinds of SNN or ANN are required. In addition, the use of only two datasets (CIFAR10 and CIFAR100) may not be sufficient to fully demonstrate the generalizability of the proposed method.


[1] Alemi, Alexander A., et al. "Deep Variational Information Bottleneck." International Conference on Learning Representations. 2017.

[2]Venkatesha, Yeshwanth, et al. "Federated learning with spiking neural networks." IEEE Transactions on Signal Processing 69 (2021): 6183-6194.

**Questions:**

How to choose the hyperparameters (e.g., $\beta$, $\epsilon$, etc.)? It seems somewhat arbitrary. There should be more justification for these choices and an exploration of how sensitive the results are to different hyperparameter settings.

---

### Note · Authors · 2024-11-21

I have read and agree with the venue's withdrawal policy on behalf of myself and my co-authors.